# Structure determination of ζ-N$_2$ from single-crystal X-ray diffraction and theoretical suggestion for the formation of amorphous nitrogen

Dominique Laniel [1] ✉, Florian Trybel [2] ✉, Andrey Aslandukov [3,4], James Spender[1], Umbertoluca Ranieri [1], Timofey Fedotenko[5], Konstantin Glazyrin [5], Eleanor Lawrence Bright[6], Stella Chariton [7], Vitali B. Prakapenka[7], Igor A. Abrikosov [2], Leonid Dubrovinsky [4] & Natalia Dubrovinskaia [2,3]

The allotropy of solid molecular nitrogen is the consequence of a complex interplay between fundamental intermolecular as well as intramolecular interactions. Understanding the underlying physical mechanisms hinges on knowledge of the crystal structures of these molecular phases. That is especially true for ζ-N$_2$, key to shed light on nitrogen's polymerization. Here, we perform single-crystal X-ray diffraction on laser-heated N$_2$ samples at 54, 63, 70 and 86 GPa and solve and refine the hitherto unknown structure of ζ-N$_2$. In its monoclinic unit cell (space group $C2/c$), 16 N$_2$ molecules are arranged in a configuration similar to that of ε-N$_2$. The structure model provides an explanation for the previously identified Raman and infrared lattice and vibrational modes of ζ-N$_2$. Density functional theory calculations give an insight into the gradual delocalization of electronic density from intramolecular bonds to intermolecular space and suggest a possible pathway towards nitrogen's polymerization.

Improving our understanding of simple diatomic molecular solids under high pressure is one of the fundamental tasks of condensed matter physics, with experimental data being widely used as benchmark tests for modern theories of the solid state. Often considered an archetypal molecular system, solid molecular nitrogen is comprised of N$_2$ molecules that exhibit the strongest homoatomic covalent bond, the shortest bond length[1] and a significant electric quadrupole moment[2], with molecules solely interacting with one another through van der Waals interactions. This appearance of simplicity is shattered as density is increased and van der Waals interactions compete with one another as well as with packing efficiency. This gives rise to a surprisingly complex phase diagram, so far composed of 12 molecular phases[3], three polymeric phases[4–7] and one amorphous[8]. Of particular interest, onwards from ~80 GPa at ambient temperature[8], ζ-N$_2$ is thought to undergo a progressive polymerization with a redistribution of its electronic density from intramolecular to intermolecular spaces,

[1]Centre for Science at Extreme Conditions and School of Physics and Astronomy, University of Edinburgh, EH9 3FD Edinburgh, UK. [2]Department of Physics, Chemistry and Biology (IFM), Linköping University, SE-581 83 Linköping, Sweden. [3]Material Physics and Technology at Extreme Conditions, Laboratory of Crystallography, University of Bayreuth, 95440 Bayreuth, Germany. [4]Bayerisches Geoinstitut, University of Bayreuth, 95440 Bayreuth, Germany. [5]Photon Science, Deutsches Elektronen-Synchrotron, Notkestrasse 85, 22607 Hamburg, Germany. [6]The European Synchrotron Radiation Facility, 38043, Grenoble Cedex 9, France. [7]Center for Advanced Radiation Sources, The University of Chicago, Chicago, IL 60637, USA. ✉e-mail: dominique.laniel@ed.ac.uk; florian.trybel@liu.se

as deduced from the onset of the redshift of its $N_2$ stretching mode[8]. This pressure-induced continuous change ultimately leads to the formation of an amorphous polymeric phase mainly composed of single-bonded nitrogen atoms[8–10]. This gradual process is of the utmost interest as it appears to be unique to molecular nitrogen: a pressure-induced polymerization does not occur in other homoatomic systems, while in $CO$[11], $CO_2$[12–14], $C_2H_2$[15], and $C_2H_4$[16] the transformation is abrupt. A full understanding of this curious chemico-physical behavior of elemental nitrogen remains to be clarified as the crystal structure of $\zeta$-$N_2$, phase stable between 60 to 110 GPa, remains unknown.

The formation of $\zeta$-$N_2$ from $\varepsilon$-$N_2$ around 60 GPa was discovered based on the appearance of new diffraction lines as well as Raman and infrared modes[17–26]. The signal of $\varepsilon$-$N_2$ observed with all three of these characterization methods is nonetheless preserved through this phase transition. This suggests an intimate structural relationship between the two phases, with $\zeta$-$N_2$ being of lower symmetry than $\varepsilon$-$N_2$. Several attempts were done to obtain a structure model of $\zeta$-$N_2$[18,19]. On the basis of powder X-ray diffraction (pXRD), an orthorhombic unit cell (space group $P222_1$, #17) was suggested, with two crystallographically unique atoms per unit cell, both on the $4e$ Wyckoff site. This model was later categorically ruled out as it implied a relatively large decrease in volume per $N_2$ molecule between $\varepsilon$-$N_2$ and $\zeta$-$N_2$–conflicting with the observed continuity of the position of X-ray diffraction lines and Raman modes at the phase transition pressure–as well as incompatibilities of the structure model with the measured vibrational modes of $\zeta$-$N_2$[17]. Indeed, very detailed spectroscopic studies at various temperatures found that $\zeta$-$N_2$ should have at least fifteen Raman modes and three IR modes, respectively with five and two stretching modes[8,22,23]. Another orthorhombic unit cell was later proposed, but without insight into the nitrogen atoms' positions[18]. Theoretical calculations were performed to shed light on this problematic, but concluded that the shallow energy landscape of molecular nitrogen precluded one candidate structure from standing out[25].

In this study, single-crystal X-ray diffraction measurements on a laser-heated sample of molecular nitrogen were performed and the structure of $\zeta$-$N_2$ was solved and refined at 63, 70 and 86 GPa. $\zeta$-$N_2$ is found to have a monoclinic unit cell and an atomic arrangement close to that of $\varepsilon$-$N_2$, mainly differing in the $N_2$ molecules'

orientations. The obtained structure provides an explanation for all previous spectroscopic observations. Density functional theory (DFT) calculations show the dynamical stability of the refined structure, in qualitative agreement with the experimental Raman spectrum and, in particular, reproduce the number of Raman active modes. Furthermore, our calculations shed light on the progressive pressure-induced delocalization of the electronic density in $\zeta$-$N_2$. A clear electronic density shift from intramolecular to intermolecular space was observed in DFT-based calculations, which can be interpreted as a precursor to the formation of single-bonded amorphous nitrogen.

## Results and discussion

Four screw-type BX90 diamond anvil cells (DACs)[27] were loaded with different laser light absorbers and molecular nitrogen, as described in full detail in the Methods section. The samples were compressed to pressures of 54 GPa (DAC1), 63 GPa (DAC2), 70 GPa (DAC3) and 86 GPa (DAC4), and heated to a maximum temperature of 2600(200) K, 3400(200) K, 2500(200) K and to an estimate temperature of 3000 K, respectively. The reaction products of molecular nitrogen with the laser absorbers are described elsewhere[28,29]. In all samples, molecular nitrogen surrounding the laser absorbers was also heated, albeit likely to a lower temperature, producing multiple tiny (< 500 nm) high-quality single-crystals on which Raman spectroscopy, powder X-ray diffraction and single-crystal X-ray diffraction (SCXRD) data were collected.

The single-crystal data collected at 54 GPa enabled to identify the expected $\varepsilon$-$N_2$ phase. Its refined lattice parameters, unit cell volume and atomic position all match those found in the literature[18,30] (Supplementary Table 1, Fig. 1a and Supplementary Fig. 1). From the pXRD and Raman spectroscopy data collected at 63, 70, and 86 GPa, the recrystallized molecular nitrogen was instead identified as $\zeta$-$N_2$ by comparing the position of the measured diffraction lines and Raman modes to those found in the literature (Fig. 1 and Supplementary Fig. 2)[8,18]. The pXRD pattern of the $\zeta$-$N_2$ allotrope is similar to that of $\varepsilon$-$N_2$[18], albeit with additional diffraction lines. The two high-temperature allotropes of molecular nitrogen, $\iota$- and $\theta$-$N_2$[20], were not detected either by pXRD, SCXRD or Raman spectroscopy. These phases have

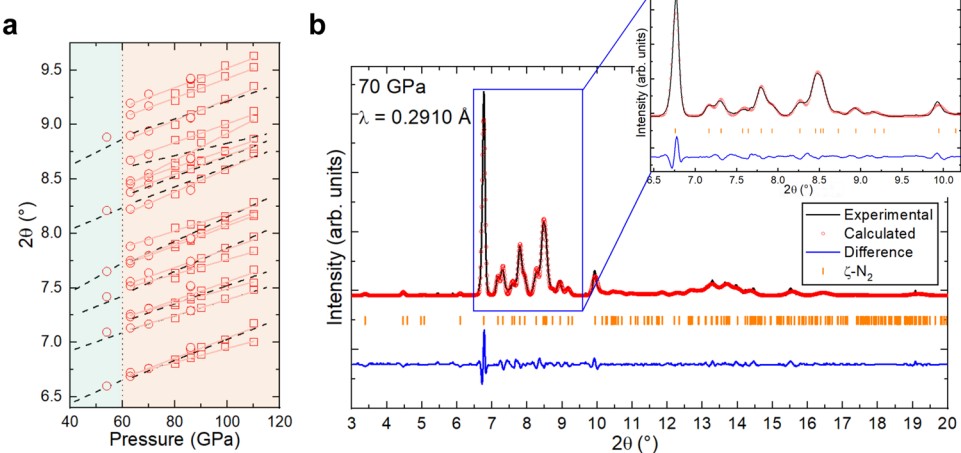

**Fig. 1 | Diffraction data analysis of molecular nitrogen between 40 and 60 GPa. a** Evolution of the position of the diffraction lines (2θ) of molecular nitrogen with pressure. Areas below and above 60 GPa, marked by cyan and orange colors, correspond to $\varepsilon$-$N_2$ and $\zeta$-$N_2$ nitrogen allotropes, respectively. The dashed black lines show the positions of the most intense diffraction lines of $\zeta$-$N_2$ as identified in Ref. 18. The empty red circles represent the positions of diffraction lines of $\zeta$-$N_2$ as observed in this study at 63, 70, and 86 GPa, each with a pressure uncertainty of ±2 GPa. The empty red squares represent the re-analyzed raw data from Ref. 18 using the Le Bail method and considering the monoclinic unit cell

determined for $\zeta$-$N_2$ in the present work on the basis of SCXRD. The continuous red lines are linear fits of the data points corresponding to a given diffraction line of $\zeta$-$N_2$. The differences between the dashed black lines and the continuous red lines originate from deconvoluting the strongly overlapping diffraction lines and including weak intensity peaks. The vertical dotted line at 60 GPa represents the phase boundary between $\varepsilon$-$N_2$ and $\zeta$-$N_2$. For the comparison, both our and literature d-spacings were recalculated to 2θ values for a wavelength λ = 0.2910 Å. **b** Le Bail analysis of the integrated 2D XRD pattern (powder pattern) of $\zeta$-$N_2$ at 70 GPa. Source data are provided as a Source Data file.

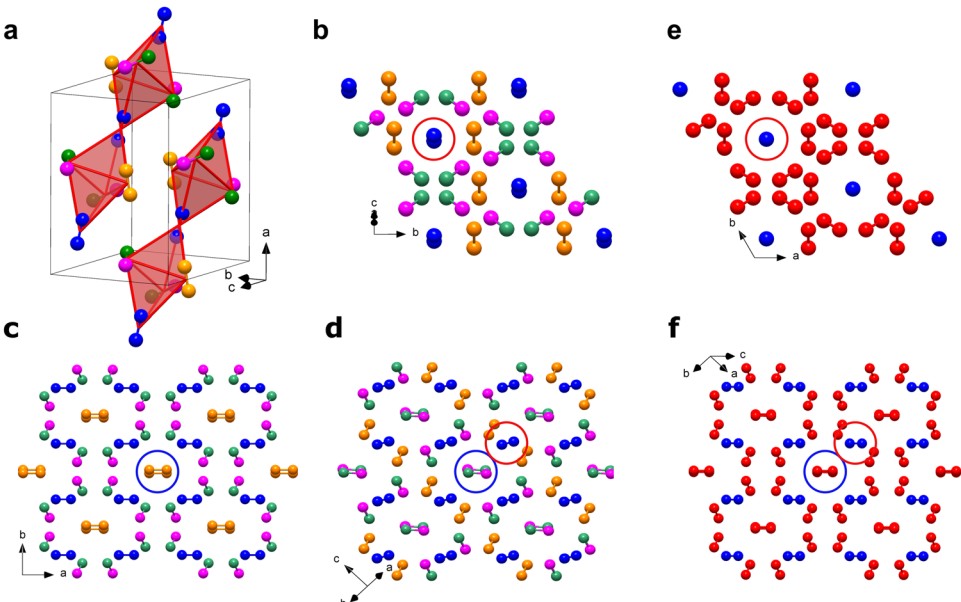

**Fig. 2 | Crystal structure of ζ-N₂ at 63 GPa and that of ε-N₂ at the same density.** The structures from (**a**) to (**d**) are of ζ-N₂ at 63 GPa and those in (**e**) and (**f**) of ε-N₂. **a** Structural motif of ζ-N₂ presented by chains of apexes-sharing triangular bipyramids (in red) aligned along the [25 0 −12] direction (a unit cell is outlined); the corners of the pyramids appear at the centers of mass of N₂ molecules. The blue, pink, orange and green spheres correspond to the N1, N2, N3, and N4 atoms; (**b**) the structure of ζ-N₂ viewed along the [25 0 −12] direction; (**c, d**) the structure of ζ-N₂ in two orientations ([0 0 1] and [−251 −195 21], respectively) helpful for visualizing its details. **e** The structure of ε-N₂ viewed along the *c* direction (the blue and red spheres represent the N1 and N2 atoms, respectively); (**f**) the structure of ε-N₂ in the orientation simplifying a comparison with the structure of ζ-N₂ shown in (**c, d**). Blue and red circles help underline the differences between ε-N₂ and ζ-N₂. The crystallographic data for ζ-N₂ at 63 GPa has been submitted under the deposition number CCDC 2237807.

already been established to be kinetically stabilized and their formation is dependent on the followed pressure–temperature path[20,31,32].

From the analysis of the SCXRD data at 63, 70, and 86 GPa (see Supplementary Table 2, 3, and 4), the structure of ζ-N₂ was determined to have a monoclinic unit cell (space group *C2/c*, #15). Supplementary Fig. 3, Supplementary Fig. 4 and Supplementary Fig. 5 show slices of the reciprocal space of the ζ-N₂ single-crystals, shown to respect the extinction conditions corresponding to the chosen space group. The lattice parameters at 63 GPa are of $a = 7.580(5)$ Å, $b = 6.635(6)$ Å, $c = 5.018(2)$ Å, and $\beta = 97.64(4)°$ ($V = 250.2(3)$ Å³). Four nitrogen atoms were found to be crystallographically distinct and occupy the 8*f* Wyckoff position, producing a total of 16 nitrogen molecules per unit cell. Three of these molecules are unique, formed by the pairs of atoms N1-N1, N3-N3, and N2-N4, featuring intramolecular distances of 1.051(6) Å, 1.016(7) Å, and 1.067(8) Å at 63 GPa, respectively, with an average value of 1.050(7) Å. This bond length value is close to the expected ~1.10 Å for triple-bonded molecular nitrogen[30].

The structure of ζ-N₂ is shown in Fig. 2 and Supplementary Fig. 6. When the molecules are linked together through their centers of mass, ζ-N₂ can be presented as an arrangement of chains of apexes-sharing and slightly distorted triangular bipyramids aligned along the [25 0 −12] direction (Fig. 2 a). The N1-N1 molecules are at the apexes of the bipyramids, while one N3-N3 and two N2-N4 molecules are at the corners of the isosceles triangular base. The three basal molecules have their centers of mass at distances of 2.719(2), 2.719(2) and 2.750(1) Å from one another, while contacts to the apexes are of 2.834(1), 2.866(1) and 2.932(2) Å. When visualized along the [25 0 −12] direction (Fig. 2 b), the structure of ζ-N₂ can be easily compared with that of ε-N₂ (Fig. 2 e). In the latter, the N1-N1 molecules (blue spheres) are all aligned along its *c* direction, whereas in the former, only the centers of mass of the N1-N1 molecules appear on the same line, so that the molecules are tilted, and one sees two blue spheres in the projection along the [25 0 −12] direction. If the structure of ζ-N₂ is viewed along its *c* (Fig. 2 c) and [−251 −195 21] directions (Fig. 2 d), one can see that the N4-N4 and N2-N3

molecules are displaced with respect to each other, if compared with the N2-N2 molecules of ε-N₂, which superimpose in the similar projection (Fig. 2 f). These tiny deviations in the structure of ζ-N₂ lead to a symmetry reduction from *R-3c* (#167) in ε-N₂ to *C2/c* (#15) in ζ-N₂.

The space group *C2/c* being a subgroup of *R-3c*, the correspondence in atomic positions between ζ-N₂ and ε-N₂ is straightforward to determine. The N1 position in ε-N₂ (12*c* Wyckoff site, blue spheres in Fig. 2e, f) directly matches the N1 position in ζ-N₂ (blue sphere in Fig. 2a–d), and the N2 position in ε-N₂ (36*f* Wyckoff site, red spheres in Fig. 2e, f) splits into the N2, N3 and N4 positions in ζ-N₂ (atoms drawn in pink, orange and green, respectively, in Fig. 2a–d). This consideration is inline with the suggestion that the ε → ζ transformation occurs through a displacive phase transition, reported previously based on the lack of a discontinuity in the evolution of the diffraction lines[17–19] and vibrons[20–24] of molecular nitrogen through the transition pressure. It is worth noting that previous work[33] had predicted a structure for ζ-N₂—based on group-subgroup related structures obtained through slight distortions of the cubic δ-N₂ parent structure—that is remarkably similar to the one here determined.

Having determined the correct unit cell of ζ-N₂, the absence of a discontinuity in the volume per N₂ molecule with pressure can also be verified. Shown in Fig. 3 is the volume per N₂ molecule at 63, 70, and 86 GPa obtained from single-crystal data. Figure 3 includes additional points at pressures of 80, 86, 90, 99, and 110 GPa, obtained from re-analyzing raw powder X-ray diffraction data from Gregoryanz et al[18] using Le Bail refinement to determine the unit cell parameters of ζ-N₂ (Supplementary Fig. 7 and Supplementary Fig. 8). Using the EoSFIT software[34], a second order Birch-Murnaghan equation of state (BM2-EOS) was used to fit the pressure-volume datapoints of ε-N₂ ($V_0 = 29.3(5)$ Å³; $K_0 = 30(2)$ GPa), and another BM2-EOS to fit the combined points of ε-N₂ and ζ-N₂ ($V_0 = 29.2(3)$ Å³; $K_0 = 30(1)$ GPa). Remarkably, the two curves are, within uncertainty, identical to one another. This perfect agreement unambiguously demonstrates the lack of a volume discontinuity between the ε- and ζ-N₂ phases.

DFT calculations performed on ζ-N₂ led to a relaxed structure which reproduces well the experimental models at 63, 70 and 86 GPa (Supplementary Tables 2, 3, and 4). The structure was found dynamically stable with its computed phonon density of states featuring no imaginary modes at 70 GPa (Supplementary Fig. 9). Calculations with the HSE functional find ζ-N₂ to have a wide bandgap of 5.3 eV at 70 GPa. The calculated band structure (Supplementary Fig. 10) predicts an indirect bandgap. The projection of the electronic density of states to the atomic orbitals (Supplementary Fig. 11) reveal that N2 and N4,

despite forming one molecule, have a slightly different contribution to the total DOS, which is in agreement with both atoms having distinct chemical environments. The static enthalpy of ζ-N₂ was also computed and compared to that of ε-N₂ (Supplementary Fig. 12) in the pressure interval from 16 to 115 GPa. The former shows a lower static enthalpy than the latter at all pressures ($\Delta H < 15$ meV/atom). Extrapolating the static enthalpy calculations leads to a transition at ~10 GPa. This is in qualitative agreement with experimental measurements at low temperature which show a ε-N₂ → ζ-N₂ transformation between ~ 10 and ~25 GPa at 30 K[22] (see Supplementary Information for details).

Previously, the most compelling evidence for ζ-N₂ came from infrared (IR) and Raman spectroscopy measurements[20–24]. In the frequency range corresponding to the N₂ molecules' vibrons, the three most recent studies[8,22,23] agreed to five and two modes detectable from Raman and IR measurements, respectively, while ten lattice modes were observed from Raman measurements[22] and one from IR studies[8]. The number of Raman modes previously reported is in agreement with the experimental data presented here for ζ-N₂ at 70 GPa (Supplementary Fig. 2).

It stands to reason that the structure model proposed for ζ-N₂ needs to account for the modes found by Raman and IR measurements. Based on group theory, the structure of ζ-N₂ can have a total of 24 Raman-active modes ($\Gamma = 12A_g + 12B_g$) and 21 IR-active modes ($\Gamma = 11A_u + 10B_u$)—so far in agreement with previous experimental evidences. To provide a more quantitative analysis, the intensity and frequency of the IR- and Raman-active modes corresponding to the structure of ζ-N₂ were calculated using DFT at 70 GPa (Supplementary Table 5). Excluding acoustic modes, 24 Raman modes were obtained: five vibron modes and 19 lattice modes. The number of vibron modes is in perfect correspondence with the experimental data, while more lattice modes are predicted than reported from measurements. However, as seen in Supplementary Table 5, most of the lattice modes are, compared to the vibrons, calculated to have a much lower intensity and to be quite close to one another in frequency—perfectly compatible with the low intensity and broad peaks detected from experiments[20,22,23] and the data in Supplementary Fig. 2. The same reasoning applies to IR modes (Supplementary Table 5), for which the 21 (three vibrons, 18 lattice modes) predicted by calculations are all found to be of very low intensity and often close in frequency. It is worth underlining that while the calculated Raman frequency of the lattice modes reproduces very well the experimental data, the agreement appears not to be as good for the vibron modes (Supplementary Fig. 2). However, these frequency differences for the vibron modes are

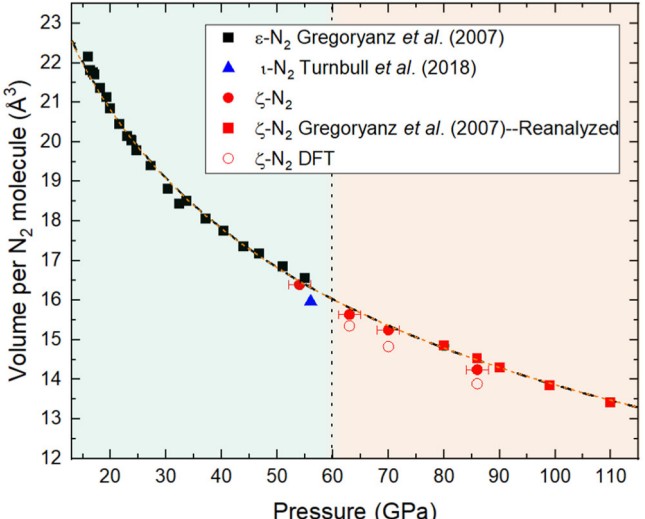

**Fig. 3 | Volume per N₂ molecule with respect to pressure for the ε-, ζ- and ι-N₂ phases.** The experimental data for ε- and ι-N₂ are according to Gregoryanz et al[18] and Turnbull et al.[3], respectively. The red circles and squares are experimental datapoints (with a pressure uncertainty of ±2 GPa), with the latter obtained from the analysis of the raw powder X-ray diffraction data presented in Ref. 18, for which no pressure uncertainty was reported. The black and the orange dashed lines represent a second order Birch–Murnaghan equation of state fit to the datapoints ε-N₂ ($V_0 = 29.3(5)$ Å³; $K_0 = 30(2)$ GPa) and the combined datapoints of ε- and ζ-N₂ ($V_0 = 29.2(3)$ Å³; $K_0 = 30(1)$ GPa), respectively. The two perfectly overlap, demonstrating the lack of a volume discontinuity between the two phases. The vertical dotted line at 60 GPa represents the phase boundary between ε-N₂ and ζ-N₂. Source data are provided as a Source Data file.

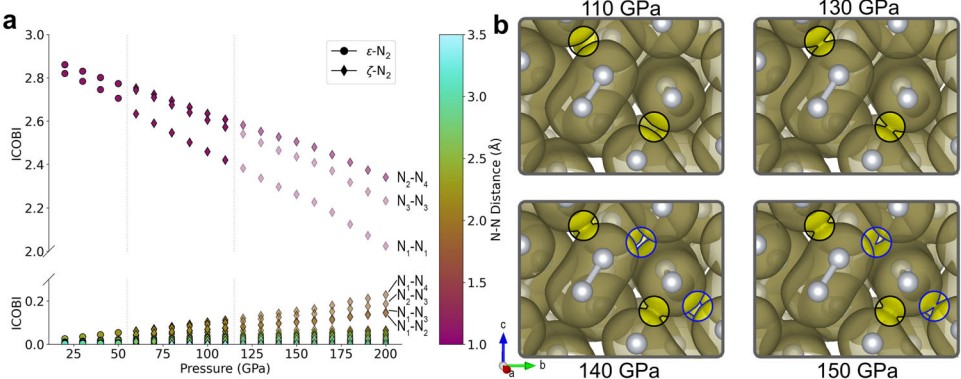

**Fig. 4 | Calculated electron density evolution in molecular nitrogen up to 200 GPa. a** Integrated crystal orbital bond index (ICOBI) values of intra (ICOBI > 2) and intermolecular (ICOBI < 0.25) N-N regions, plotted for both ε- and ζ-N₂. Some ICOBI values are labeled with the nitrogen atoms' pair they represent. The structure of ζ-N₂ is used past 115 GPa, as the structure of κ-N₂, stable from this pressure on, is unknown although expected to be very similar to that of ζ-N₂[18]. The dashed vertical lines denote the transition pressure between the ε → ζ and ζ → κ phases.

**b** Electron localization function (ELF) isosurfaces of 0.5 plotted at pressures of 110, 130, 140, and 150 GPa. Black and blue circles were added to highlight the regions where N₂ molecules' ELF isosurfaces start to connect at 130 GPa and at 150 GPa, respectively, and portions outside of these circles were grayed out for clarity. Source data are provided as a Source Data file and version for (**b**) without the grayed-out areas can be found in Fig. S15.

only of between 0.2 and 1%−values which are in fact quite reasonable and in line with what can be expected from theoretical calculations of such systems[35–40].

The unambiguous structure solution of ζ-$N_2$ obtained in this work, in combination with DFT calculations, allows us to provide a qualitative analysis of the gradual pressure-induced electronic density delocalization in molecular nitrogen and thus shed light on the unique attribute of nitrogen−its progressive molecular-to-polymeric phase transition. As the structure of κ-$N_2$, reported to form above 115 GPa[18], is still unknown, DFT calculations were performed using the structure of ζ-$N_2$[18] up to 200 GPa. Since κ-$N_2$ is expected to be structurally very similar to ζ-$N_2$[18], this allows for a qualitatively sound analysis.

The integrated crystal orbital bond index (ICOBI)[41]−a direct measure of the bond order−was computed between 20 and 200 GPa (Fig. 4 a), for both intramolecular and intermolecular nitrogen contacts. Interestingly, the intramolecular ICOBI decreases almost linearly with pressure, beginning at 20 GPa in the ε-$N_2$ phase, concomitantly with an ICOBI increase between $N_2$ units (Fig. 4 a). The intramolecular bond with the lowest ICOBI value is that of the N1-N1 molecule (Fig. 4 a): precisely the one for which the vibron peak begins to redshift at 80 GPa[8,24]. The intramolecular ICOBI at 200 GPa are between 2.0 and 2.4, while the ICOBI corresponding to the intermolecular regions is substantially higher for four pairs of atoms: ~0.230 (N1-N4), ~0.190 (N2-N3), ~0.148 (N1-N3) and ~0.144 (N1-N2), respectively at distances of 1.83, 1.85 and 1.89 Å from one another. Again, one can note the electronic dissimilarities between the atoms of the N2-N4 molecule. The slope of the intramolecular ICOBI with respect to pressure changes for all three distinct molecules between 90 and 110 GPa (Supplementary Fig. 13); which denotes the acceleration of the intra-to-intermolecular electronic delocalization at higher pressures.

To better understand the redistribution of electronic density, the electron localization function (ELF) of ζ-$N_2$ was calculated at pressures between 110 to 150 GPa. From the ELF computations (Fig. 4b), isosurfaces of 0.5−value defined as that of a free electron gas[42]−were drawn for structures in this pressure range. At 130 GPa, ELF isosurfaces start to connect in the intermolecular space between the N1 and N4 atoms (1.96 Å apart), each, respectively, part of the N1-N1 and N2-N4 molecules, altogether forming three-molecule-long units (i.e. two N1-N1 and one N2-N4). This N1 to N4 intermolecular region is the one with the highest calculated ICOBI value. At a pressure of 150 GPa, this ELF bridge extends to the N3-N3 molecule, as seen in Fig. 4b), with the N1 atom then forming three contacts with ELF values greater than 0.5. This results in a 1D percolation of ELF-linked molecules in ζ-$N_2$ which, when visualized along the *c*-axis, shows structural similarities to cg-N (Supplementary Fig. 14). This network could act as a seed for nitrogen's polymerization, thus providing a tentative path for this unique transformation. Given the mild volume underestimation of theoretical calculations compared to the experimentally measured volumes−and not accounting for the κ-$N_2$ phase−the pressure at which this phenomenon occurs is likely higher. The performed static DFT calculations put on a solid footing a progressive pressure-induced polymerization and therefore establish a potential underpinning mechanism for the formation of single-bonded amorphous nitrogen. Extended molecular dynamics simulations at finite temperatures, which capture thermal effects as well as possible intermediate phases[43,44] during the amorphization, would be the next step in order to study this phenomenon in more detail.

The in-situ single-crystal X-ray diffraction data allowed solving and refining the structure of ζ-$N_2$ at 63, 70, and 86 GPa. ζ-$N_2$ has pronounced similarities with ε-$N_2$ having analogous arrangements of molecules but, as permitted by its lower symmetry, has subtle differences in the molecules' orientation. This provides an explanation for the observed continuity of the X-ray diffraction lines as well as Raman and infrared modes across the ε → ζ phase transition, as well as the appearance of new signals. Moreover, the determined structure of ζ-$N_2$

fully explains the previously collected detailed spectroscopic data with regard to the known number of lattice and vibron modes.

Theoretical calculations show a progressive intra-to-intermolecular electronic delocalization occurring in molecular nitrogen, also identifying a mild acceleration of this phenomenon near 100 GPa. The proximity of the $N_2$ molecules at 150 GPa−and in particular those identified by ELF to agglomerate−could act as a seed for polymerization and suggests a natural transformational mechanism towards amorphous single-bonded nitrogen. This study thus brings us one step closer to a full understanding of this deceivingly simple element under pressure.

## Methods
### Experimental
**DAC preparation and laser-heating.** Four BX90-type screw-driven diamond anvil cells (DACs)[27] were equipped with 250 μm culet diamond anvils. Rhenium gaskets with an initial thickness of 200 μm were indented down to ~25 μm and a hole of ~120 μm in diameter was laser-drilled at the center of the indentations. The four DACs were loaded with YAG laser light absorbers, namely potassium azide ($KN_3$, Sigma Aldrich, 99.9% purity) in DAC2, molecular nitrogen loaded as a gas at 1200 bars[45] along with carbon tetraiodide ($CI_4$, Sigma Aldrich 97% purity) in DAC1, tetracyaoethylene (TCNE, $C_6N_4$, Alfa Aesar, 98% purity) in DAC3, and a boron-doped carbon disk in DAC4. The in-situ pressure was determined using the equation of state of rhenium[46] and verified using the first-order Raman mode of the stressed diamond anvils[47]. Double-sided sample laser heating was performed at the Bayerisches Geoinstitut[48] and at the P02.2 beamline. Temperatures were measured with an accuracy of ±200 K, using the thermoemission produced by the laser-heated samples[48].

**Synchrotron X-ray diffraction.** Synchrotron X-ray diffraction measurements of the compressed samples were performed at the P02.2 beamline ($\lambda = 0.2890$ Å or $\lambda = 0.2910$ Å) of PETRA III, at the ID11 beamline ($\lambda = 0.2846$ Å) of the ESRF and at the GSECARS beamline ($\lambda = 0.2950$ Å) of the APS. In order to determine the position where the single-crystal X-ray diffraction (SCXRD) data would be collected, the sample cavity was fully mapped by X-ray diffraction. The sample positions displaying the most and the strongest single-crystal reflections belonging to ζ-$N_2$ were chosen for the acquisition of single-crystal data in step-scans of 0.5° from −38° to +38°, resulting in 152 diffraction images per dataset.

The CrysAlisPro software[49] was utilized for the analysis of the single crystal data. Previously to the single-crystal data acquisitions on ζ-$N_2$, data had been collected on a single-crystal standard in order to obtain in CrysAlis$^{Pro}$ the calibrated diffractometer's model. For this, a standard orthoenstatite single-crystal loaded in a DAC, ($Mg_{1.93}Fe_{0.06}$)($Si_{1.93}$,$Al_{0.07}$)$O_6$, space group *Pbca* with lattice parameter $a = 8.8117(2)$ Å, $b = 5.18320(10)$ Å, and $c = 18.2391(3)$ Å, was employed.

Once the calibration has been done, the first step of the data analysis consists of converting the 152 diffraction images constituting a dataset to the ESPERANTO format−i.e. the format used by CrysAlis$^{Pro}$. Then, the peak hunting command (3D peak search option) of CrysAlis$^{Pro}$ is used, which searches through all 152 diffraction images to find diffraction spots. The position of these diffraction spots in the reciprocal space is saved to a .tabbin file, which is given to the DAFi software[50]. This software searches through these reflections and identifies groups of reflections belonging to individual single-crystal domains. The output of DAFi is provided back to CrysAlis$^{Pro}$, in which the Auto unit cell finding in the shown peaks command is ran to determine the unit cell and orientation matrix of all single-crystals for which groups of reflections were found by DAFi.

Having established the crystal's unit cell, the reflections' intensity need to be extracted to later be used to determine the atomic species and position. This is done through CrysAlis$^{Pro}$'s data reduction (or

integration) procedure. In essence, CrysAlis$^{Pro}$ goes again through all 152 diffraction images and, using user-provided parameters such as the opening angle of the DAC, background evaluation mode, integration box size, reflection profile fitting mode as well as the known unit cell parameters and orientation matrix, the intensity of each reflection of a given single-crystal is obtained. CrysAlis$^{Pro}$ also applies frame scaling and absorption corrections. Systematic absences are analysed and a space group suggested. After the data integration is process, an $R_{int}$ value (see Supplementary Table 1, 2, 3, and 4) is provided, which is a figure of merit of the integration. $R_{int}$ values below 10% are considered as good and likely to be sufficient to obtain a structure model.

The *hkl* file, a standard format listing the observed reflections by their *hkl* index and intensity, is produced by CrysAlisPro at the end of the integration process. Normal structure solving and refinement softwares can be employed using this file. In our case, the JANA2006 software[51] was used, along with the SHELXT and SHELXL softwares[52] for structure solving and refinement, respectively. Reflections overlapping with parasitic signal, either from the diamond anvils or distinct single-crystals, were omitted from the refinement. This single-crystal X-ray diffraction approach for polycrystalline samples was successfully employed by other independent research groups[47–49].

Powder X-ray diffraction measurements were also performed and the obtained data was integrated with the Dioptas software[53] and Le Bail refinements completed using FullProf[54].

**Raman spectroscopy.** Confocal Raman spectroscopy measurements were performed at the Centre for Science at Extreme Conditions (CSEC). The measurements were conducted using 532 nm excitation wavelength via a custom-built micro-focused Raman system equipped with a 20x Mitutoyo objective, a spectrometer with 1800 lines/mm grating and a CCD array detector. A laser output power of 63 mW was used.

**Computational details**
Kohn-Sham density functional theory (DFT) based calculations were performed with the Quantum Espresso package[55–57] using the projector augmented wave method[58]. We used the generalized gradient approximation by Perdew-Burke-Ernzerhof (PBE) for exchange and correlation[59] with a projector augmented wave potential file, where the 1 s electrons are treated as scalar-relativistic core state. We include van der Waals corrections following the approach by Grimme et al. as implemented in Quantum Espresso[60]. Convergence tests with a threshold of 1 meV per atom in energy and 0.1 meV/Å per atom for forces led to a Monkhorst-Pack[61] *k*-point grid of 6x7x9 for the conventional cell and an energy cutoff for the plane wave expansion of 80 Ry. We performed variable cell relaxations (lattice parameters and atomic positions) on the experimental structures to optimize the atomic coordinates and the cell vectors until the total forces were smaller than $10^{-4}$ eV/Å per atom and the deviation from the experimental pressure was below 0.1 GPa.

**Phonon calculations.** Harmonic phonons were calculated with Phonopy[62] in 432 atom supercells with adjusted *k*-points. The position of the phonon modes at the Γ-point and Raman activities were furthermore calculated using *ph.x* within the Quantum Espresso package using Optimized Norm-Conserving Vanderbilt (ONCV) potentials[63] with an adjusted cutoff of 120 Ry. Frequencies are in very good agreement between the two calculation methods.

**Electronic structure calculations.** Electronic structure calculations are performed with a higher *k*-point resolution of 12x14x18. The same parameters are used to calculate electron localization functions (ELFs) as well as the crystal orbital bond index[41] via the LOBSTER code[64]. ELFs are plotted with VESTA3[65].

Additional calculations for the band gap are performed with the Heyd–Scuseria–Ernzerhof (HSE) hybrid functional[66] with the standard screening parameter, a *k*-grid of 8x8x8 and a *q*-grid of 4x4x4.

## Data availability
Crystallographic data for the structures reported in this Article have been deposited at the Cambridge Crystallographic Data Centre, under deposition numbers CCDC 2237807-2237808, 2261391, and 2280044. Copies of the data can be obtained free of charge via https://www.ccdc.cam.ac.uk/structures/. Source data for Fig. 1 a, b; Fig. 3; Fig. 4; Supplementary Fig. 2a, b; Supplementary Fig. 7a, b, c, d, e; Supplementary Fig. 8; Supplementary Fig. 9, Supplementary Fig. 10, Supplementary Fig. 11, Supplementary Fig. 12 and Supplementary Fig. 13 are provided with this paper. The data for Fig. 4 b are provided as a zipped file containing the "cube" datatype. The Crystallographic Information Files (CIFs) for ε-N$_2$ at 54 GPa as well as ζ-N$_2$ at 63, 70, and 86 GPa can be found in Supplementary Data 1. Supplementary Data 2 contains the Electron Localization Functions (ELFs) of ζ-N$_2$ at 110, 130, 140 and 150 GPa. The corresponding authors can be contacted for any requests. Source data are provided with this paper.

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

## Acknowledgements

The authors thank E. Gregoryanz for sharing with us his raw powder X-ray diffraction data on ζ-N$_2$ and F. Tasnádi for helpful discussions. The authors acknowledge the Deutsches Elektronen-Synchrotron (DESY, PETRA III) and the European Synchrotron Radiation Facility (ESRF) for provision of beamtime at the P02.2 and the ID11 beamline, respectively. Portions of this work were also performed at GeoSoilEnviroCARS (The University of Chicago, Sector 13), Advanced Photon Source (APS), Argonne National Laboratory. GeoSoilEnviroCARS is supported by the National Science Foundation—Earth Sciences (EAR—1634415). This research used resources of the Advanced Photon Source, a U.S. Department of Energy (DOE) Office of Science User Facility operated for the DOE Office of Science by Argonne National Laboratory under Contract No. DE-AC02-06CH11357. D.L. thanks the UKRI Future Leaders Fellowship (MR/V025724/1) for financial support. N.D. and L.D. thank the Federal Ministry of Education and Research, Germany (BMBF, grant no. 05K19WC1) and the Deutsche Forschungsgemeinschaft (DFG; projects DU 954–11/1, DU 393–9/2, and DU 393–13/1) for financial support. N.D. and I.A.A. also thank the Swedish Government Strategic Research Area in Materials Science on Functional Materials at Linköping University (Faculty Grant SFO-Mat-LiU No. 2009 00971). I.A.A. and F.T. are supported by the Knut and Alice Wallenberg Foundation (Wallenberg Scholar grant no. KAW-2018.0194). I.A.A. acknowledges support from the Swedish Research Council (VR) Grant No. 2019-05600. The computations were enabled by resources provided by the Swedish National Infrastructure for Computing (SNIC) at PDC though grant SNIC 2022/1-6 partially funded by the Swedish Research Council through grant agreement no. 2018-05973. We acknowledge the National Academic Infrastructure for Supercomputing in Sweden (NAISS) and the Swedish National Infrastructure for Computing (SNIC) partially funded by the Swedish Research Council through grant agreement no. 2022-06725 and no. 2018-05973 for awarding access to the LUMI supercomputer, owned by the EuroHPC Joint Undertaking, hosted by CSC (Finland) and the LUMI consortium, through grant SNIC 2022/13-3 & SNIC 2022/21-10. For the purpose of open access, the author has applied a Creative Commons Attribution (CC BY) licence to any Author Accepted Manuscript version arising from this submission.

## Author contributions

D.L. designed the work. D.L. prepared the high-pressure experiments. D.L., A.A., L.D., T.F., K.G., J.S., U.R., E.L.B., S.C., and V.B.P. performed the synchrotron X-ray diffraction experiments. D.L. processed the synchrotron X-ray diffraction data. D.L. collected and analyzed the Raman data. F.T. and I.A.A. performed the theoretical calculations. D.L. F.T., L.D., and N.D. contextualized the data interpretation. D.L., F.T., L.D., N.D., and I.A.A. prepared the first draft of the manuscript with contributions from all other authors. All the authors commented on successive drafts and have given approval to the final version of the manuscript.

## Competing interests

The authors declare no competing interests.
