## [Peer Review File · Nature Communications]

REVIEWER COMMENTS

Reviewer #1 (Remarks to the Author):

In the submitted manuscript, the authors intend to solve the hitherto unknown crystal structure of ζ -N₂. The single-crystal XRD results support a monoclinic structure, the validity of which was then further verified by comparison of the experimental and simulated Raman and infrared modes at the phase transition pressure. Also, DFT calculations basing on the new structure provide an insight into the polymerization of nitrogen at high pressure. The manuscript is well written and contains enough new results, I recommend the publication after considering the following issues.

The manuscript presents the XRD results at only two selected pressures of 63 and 70 GPa, the data obtained is not enough to show the real evolution of the main peak positions. For example, the red circles presented in Fig. 1(a) show a decreased trend around 2θ of 8.25 (8.7), violating that obtained in the previous results. I suggest the authors consider more pressure points in the XRD experiments. The authors claim that the simulated Raman modes are in good qualitative agreement with the experimental data, however, such a good agreement is not supported in the high frequency region, as shown in Fig.S1 (b).

Reviewer #2 (Remarks to the Author):

Understanding the structural and physical properties of simple diatomic molecular solids in a wide range of P-T space is of fundamental interest to solid state physics. This manuscript aims to provide an unambiguous structural solution of ζ -N₂ which would shed light to the polymerization of solid molecular nitrogen. The goal of the manuscript, if achieved, is fundamentally important. However, the experimental results and theoretical calculations of the claimed ζ -N₂ are not sound. This study does not present the level of novelty that is suitable for Nature Communications. I cannot recommend the publication in Nature communications.

Experimentally, this manuscript simply presented some Raman and pXRD data on two pressures, i.e., 63 and 70 GPa, of the ζ -N₂ phase. The most important single-crystal XRD data as well as how the authors solve the unambiguous structure using the collected single-crystal XRD data is not present in the manuscript, except referring to a paper and presenting the crystallographic data in table S1 and S2. Simply looking at Fig. 3, these two data points of ζ -N₂ seem to suggest a molecular volume discontinuity through the ϵ - ζ N₂ phase transition, which is against to what the authors presented in the manuscript text as well as literature studies. The dashed line (the molecular volume change as a function of pressure) is very misleading. The manuscript refers heavily to previous experimental data reported by others, which technically is not wrong and should be encouraged. However, conclusions based on two experimental data points are not technically sound. For example, what are the structural details of the ϵ -N₂ before the phase transformation and how the ζ -N₂ phase evolve structurally to higher pressures? Without these experimental characterization, the implication of the ϵ - ζ N₂ phase transition as well as the formation of amorphous nitrogen is not convincing.

The DFT results also raise many questions. For instance, why are the DFT calculated molecular volumes much smaller than experimental values for 63 and 70 GPa? Calculated static enthalpy difference between ζ - and ϵ -N₂ seems to suggest that ζ -N₂ is always more stable than ϵ -N₂ over a very wide pressure range, from 20 GPa to >110 GPa. This contradicts with experimental observations, and why?

A minor note: "The blue, pink, green and orange spheres correspond to the N1, N2, N3 and N4 atoms" in Fig. 2 and the text is not right. N3 atom should be orange based on authors' description.

In sum, the goal of the manuscript is scientifically important. But the study does not present sophisticated, convincing experimental or theoretical characterization to support the conclusion.

Reviewer #3 (Remarks to the Author):

Using laser-heated screw-type BX90 diamond anvil cells Laniel and coauthors synthesized and identified ζ -N₂ with XRD and Raman spectroscopy. They experimentally determined the crystal structure of ζ -N₂ at 63 and 70 GPa and theoretically extended the predicted structure to 200 GPa. The structure determination of ζ -N₂ and full explanation of the previously collected detailed Raman spectroscopic data with regards to the known number of lattice and vibron modes represent an important progress. The theoretical approach that treated ζ -N₂ and κ -N₂ as minor distortions of the ϵ -N₂ and focused on the general linear trend helped to get the big picture of progressive intra-to-intermolecular electronic delocalization and identification of ELF agglomeration occurring in molecular nitrogen. The paper is publishable in Nature Communications with the follow comment.

The central contribution and strength of the present work is the single-crystal XRD (SCXRD) of ζ -N₂ that distinguish itself from many previous works since 2006 that relied on powder XRD (pXRD) data. However, the present main text only shows powder data in Fig. 1, and leave SCXRD in the Supplemental Materials which only show the processed final answer in S1 and S2. Without any detailed data and discussion on the quality of the single crystals, it is difficult to assess the experiment

Response to referees' comments:

To better structure our response to the referees' reports, each of their comments has been copied and our response to each of these comments is written in blue. When changes were made to the manuscript, the original statement from our paper is copied and followed by the updated text. The same changes are also highlighted in the marked up manuscript file.

Report of Reviewer #1

Referees' comments:

Referee #1 (Remarks to the Author):

In the submitted manuscript, the authors intend to solve the hitherto unknown crystal structure of ζ -N₂. The single-crystal XRD results support a monoclinic structure, the validity of which was then further verified by comparison of the experimental and simulated Raman and infrared modes at the phase transition pressure. Also, DFT calculations basing on the new structure provide an insight into the polymerization of nitrogen at high pressure. The manuscript is well written and contains enough new results, I recommend the publication after considering the following issues. We thank the Reviewer for his/her very positive assessment of our manuscript.

The manuscript presents the XRD results at only two selected pressures of 63 and 70 GPa, the data obtained is not enough to show the real evolution of the main peak positions. For example, the red circles presented in Fig. 1(a) show a decreased trend around 2θ of 8.25 (8.7), violating that obtained in the previous results. I suggest the authors consider more pressure points in the XRD experiments.

We agree with the Reviewer and have followed his/her suggestion to increase the number of data points. Therefore, we have performed one more experiment and obtained single-crystal XRD data for ζ -N₂ at 86 GPa. Additionally, we requested the raw powder XRD data from the authors of Ref. 18, who provided us with their raw diffraction data at pressures of 80, 86, 90, 99 and 110 GPa. We used these powder XRD data for the Le Bail analysis with the now known monoclinic unit cell of ζ -N₂ (see new Figure S6). The refinement of the unit cell parameters of ζ -N₂ for the five pressure points from Ref. 18 (80, 86, 90, 99 and 110 GPa), in addition to our own three experimental data points (63, 70, and 86 GPa), enabled us to show confidently the real evolution of the main peak positions, as suggested by the Reviewer (see Figure 1 below; Figure 1b in the revised manuscript). We also point out with these many additional P-V points were also added to Figure 3 and enabled us to obtain the bulk modulus of ζ -N₂.

Figure 1: Evolution of the position of the diffraction lines (2θ) of molecular nitrogen with pressure. Areas below and above 60 GPa, marked by pink and light green colors, correspond to the ϵ -N₂ and ζ -N₂ molecular nitrogen allotropes, respectively. The dashed lines show the positions of the most intense diffraction lines of ζ -N₂ as identified in Ref. 18. The empty red circles represent the positions of the diffraction lines of ζ -N₂ as observed in this study at 63, 70 and 86 GPa in the 2θ range reported by Gregoryanz et al.¹⁸. The empty red squares represent the re-analysed raw data from Ref. 18 using the Le Bail method and considering the monoclinic unit cell determined for ζ -N₂ in the present work on the basis of single-crystal XRD. The continuous red lines are linear fits of the data points corresponding to a given diffraction line of ζ -N₂. The differences between the dashed black lines and the continuous red lines originate from deconvoluting the strongly overlapping diffraction lines and including weak intensity peaks. For the comparison, both our and literature d-spacings were recalculated to 2θ values for a wavelength $\lambda = 0.2910$ Å.

The authors claim that the simulated Raman modes are in good qualitative agreement with the experimental data, however, such a good agreement is not supported in the high frequency region, as shown in Fig.S1 (b).

We agree with the Reviewer that the agreement for the modes at high frequencies is not perfect. The wording was changed, and the word “good” removed.

However, we still want to highlight that the frequency difference between the calculated and the experimental modes are of 0.71% (2.9%), 0.8% (1.5%), 0.6% (1.7%), 1.0% (1.3%) and 0.2% (2.0%) with (without) an overall shift of 2.3% that is discussed in the caption of Figure S2. This mild difference is actually quite common for molecular phases and, in particular, for nitrogen-bearing phases, as reported in the following papers (We have noted this fact in the revised manuscript and have added these six references):

(1) Ma et al. (2007) Phys. Rev. B 76, 064101 “Structure of the metallic ζ -phase of oxygen and isosymmetric nature of the ϵ - ζ phase transition: Ab initio simulations”

- (2) Niwa et al. (2021) *J Raman Spectrosc.* 52:1064–1072. “Nitriding synthesis and structural change of phosphorus nitrides at high pressures”
- (3) Bhadram et al. (2018) *Phys. Rev. Mat* 2, 011602(R) “Semiconducting cubic titanium nitride in the Th3P4 structure”
- (4) Niwa et al. (2014) *Inorg. Chem.* 53, 697–699 “High Pressure Synthesis of Marcasite-Type Rhodium Pernitride”
- (5) Salke et al. (2021) *Phys. Rev. Lett.* 126, 065702. “Tungsten Hexanitride with Single-Bonded Armchairlike Hexazine Structure at High Pressure”
- (6) Ceppatelli et al (2022) *Inorg. Chem.* 61, 12165-12180. “High-Pressure and High-Temperature Chemistry of Phosphorus and Nitrogen: Synthesis and Characterization of α - and γ -P3N5”

Report of Reviewer #2

Understanding the structural and physical properties of simple diatomic molecular solids in a wide range of P-T space is of fundamental interest to solid state physics. This manuscript aims to provide an unambiguous structural solution of ζ -N₂ which would shed light to the polymerization of solid molecular nitrogen. The goal of the manuscript, if achieved, is fundamentally important. However, the experimental results and theoretical calculations of the claimed ζ -N₂ are not sound. This study does not present the level of novelty that is suitable for Nature Communications. I cannot recommend the publication in Nature communications.

Experimentally, this manuscript simply presented some Raman and pXRD data on two pressures, i.e., 63 and 70 GPa, of the ζ -N₂ phase. The most important single-crystal XRD data as well as how the authors solve the unambiguous structure using the collected single-crystal XRD data is not present in the manuscript, except referring to a paper and presenting the crystallographic data in table S1 and S2.

First, we want to point out that in the submitted version of the manuscript, the single-crystal XRD data was presented in accordance with the Nature Communications policy: Tables S1 and S2 with the crystallographic information were accompanied by Crystal Information Files (CIFs) with the structure factors included, and these CIFs had been checked with IUCr's CheckCIF routine and its output, with A- and B-alerts explained, provided. Moreover, these CIFs were submitted to the Cambridge Structural Database and their identification number supplied (i.e. CSD 2237807 and CSD 2237808). We also note that these CIFs were made available to the Reviewer so that she/he can check herself/himself the data and arrive to by her/his own means to the structure model.

Regarding the description of the DAC single-crystal X-ray diffraction approach that we use, we point out that it is now fairly standard, with independent research groups employing it (see provided Refs. 52-54). The extensive description of the methodology was published by our group six years ago in Nature Communications (doi.org/10.1038/ncomms15960), and since has become state-of-the-art and implemented at high-pressure beam-lines of all major synchrotrons (ESRF, APS, and PETRA III).

In the revised version of the manuscript, we have additionally provided slices of the reciprocal space of ϵ -N₂ at 54 GPa (Figure S1) along with those of ζ -N₂ at 63, 70 and 86 GPa (Figure S3, Figure S4 and Figure S5). This allows to visually evaluate the quality of the single-crystal XRD data.

To ensure that any reader still new to this method finds all information, the following text was added:

“Synchrotron X-ray diffraction measurements of the compressed samples were performed at the P02.2 beamline ($\lambda = 0.2890 \text{ \AA}$ or $\lambda = 0.2910 \text{ \AA}$) of PETRA III, at the ID11 beamline ($\lambda = 0.2846 \text{ \AA}$) of the ESRF and at the GSECARS beamline ($\lambda = 0.2950 \text{ \AA}$) of the APS. In order to determine the position where the single-crystal X-ray diffraction (SCXRD) data would be collected, the sample cavity was fully mapped by X-ray diffraction. The sample positions displaying the most and the strongest single-crystal reflections belonging to $\zeta\text{-N}_2$ were chosen for the acquisition of single-crystal data in step-scans of 0.5° from -38° to $+38^\circ$, resulting in 152 diffraction images per dataset.

The CrysAlisPro software⁴⁶ was utilized for the analysis of the single crystal data. Previously to the single-crystal data acquisitions on $\zeta\text{-N}_2$, data had been collected a single-crystal standard in order to obtain in CrysAlisPro the calibrated diffractometer’s model. For this, a standard orthoenstatite single-crystal loaded in a DAC, $(\text{Mg}_{1.93}\text{Fe}_{0.06})(\text{Si}_{1.93},\text{Al}_{0.07})\text{O}_6$, space group *Pbca* with lattice parameter $a = 8.8117(2) \text{ \AA}$, $b = 5.18320(10) \text{ \AA}$, and $c = 18.2391(3) \text{ \AA}$, was employed. Once the calibration has been done, the first step of the data analysis consists of converting the 152 diffraction images constituting a dataset to the “ESPERANTO” format—*i.e.* the format used by CrysAlisPro. Then, the “peak hunting” command (“3D peak search” option) of CrysAlisPro is used, which searches through all 152 diffraction images to find diffraction spots. The position of these diffraction spots in the reciprocal space is saved to a “.tabbin” file, which is given to the DAFi software⁴⁷. This software searches through these reflections and identifies groups of reflections belonging to individual single-crystal domains. The output of DAFi is provided back to CrysAlisPro, in which the “Auto unit cell finding in shown peaks” command is ran to determine the unit cell and orientation matrix of all single-crystals for which groups of reflections were found by DAFi.

Having established the crystal’s unit cell, the reflections’ intensity need to be extracted to later be used to determine the atomic species and position. This is done through CrysAlisPro’s data reduction (or “integration”) procedure. In essence, CrysAlisPro goes again through all 152 diffraction images and, using user provided parameters such as the opening angle of the DAC, background evaluation mode, integration box size, reflection profile fitting mode as well as the known unit cell parameters and orientation matrix, the intensity of each reflection of a given single-crystal is obtained. CrysAlisPro also applies frame scaling and absorption corrections. Systematic absences are analysed and a space group suggested. After the data integration process, an R_{int} value (see Table S1, S2, S3 and S4) is provided, which is a figure of merit of the integration. R_{int} values below 10% are considered as good and likely to be sufficient to obtain a structure model.

The *hkl* file, a standard format listing the observed reflections by their *hkl* index and intensity, is produced by CrysAlisPro at the end of the integration process. Normal structure solving and refinement softwares can be used using this file. In our case, the JANA2006 software⁴⁸ was employed, along with the SHELXT and SHELXL software⁴⁹ for structure solving and refinement,

respectively. Reflections overlapping with parasitic signal, either from the diamond anvils or distinct single-crystals, were omitted from the refinement.”

Simply looking at Fig. 3, these two data points of ζ -N₂ seem to suggest a molecular volume discontinuity through the ϵ - ζ N₂ phase transition, which is against to what the authors presented in the manuscript text as well as literature studies. The dashed line (the molecular volume change as a function of pressure) is very misleading.

We are sorry that Reviewer #2 found Fig. 3 misleading; corrections have now been made. In particular, in the re-submitted version of the manuscript, the arguments for a lack of a volume discontinuity between the ϵ and ζ -N₂ phases are further strengthened. Indeed, we have obtained a new single-crystal data point at 86 GPa. Moreover, we also requested and acquired the raw powder X-ray diffraction data presented in Ref. 18 from one of its authors and re-analyzed it. Through Le Bail refinements of the monoclinic unit cell of ζ -N₂ (as we determined in this work from single-crystal XRD) at five additional pressures, 80, 86, 90, 99 and 110 GPa (see Figure S6), we got an additional five data points from the powder XRD data. These datapoints are now included in Figure 3 (see below), and were employed to fit both the ϵ - and ζ -N₂ datapoints with a single second order Birch-Murnaghan equation of state (orange curve). We see that this curve perfectly overlaps with the (black) curve obtained from the equation of state of solely the ϵ -N₂ datapoints. This unambiguously shows that there is no volume discontinuity between the two phases. The text in the manuscript was amended to reflect this.

Figure 3: Volume per N₂ molecule with respect to pressure for the ϵ -, ζ - and ι -N₂ phases. The experimental data for ϵ - and ι -N₂ are according to Gregoryanz et al.¹⁸ and Turnbull et al.³, respectively. The red circles and squares are experimental datapoints, with the latter obtained from the analysis of the raw powder X-ray diffraction data presented in Ref. 18, for which no pressure uncertainty was reported. The black and the orange dashed lines represent a second order Birch-Murnaghan equation of state fit to the datapoints ϵ -N₂ ($V_0 = 29.3(5) \text{ \AA}^3$; $K_0 = 30(2) \text{ GPa}$) and the combined datapoints of ϵ - and ζ -N₂ ($V_0 = 29.2(3) \text{ \AA}^3$; $K_0 = 30(1) \text{ GPa}$), respectively. The two perfectly overlap, demonstrating the lack of a volume discontinuity between the two phases. The dashed vertical line at 60 GPa represents the pressure of the ϵ - to ζ -N₂ phase transition.

The manuscript refers heavily to previous experimental data reported by others, which technically is not wrong and should be encouraged. However, conclusions based on two experimental data points are not technically sound.

To clarify, we would like to emphasize that the fundamental novelty of our work is the determination of the crystal structure of ζ -N₂, solved based on hundreds of single-crystal reflections. However, we understand the Reviewer's concerns that for an equation of state, or to understand a phase's behavior with pressure, two datapoints are very few. As such, we re-analysed the powder diffraction patterns of a previous study, adding another five P-V datapoints to the ζ -N₂ equation of state (see Figure 3 and Figure S6). All newly added datapoints are in agreement with our previous assessment. Another new datapoint, this one obtained from single-crystal X-ray diffraction, was also added.

For example, what are the structural details of the ϵ -N₂ before the phase transformation and how the ζ -N₂ phase evolve structurally to higher pressures?

The structure of ϵ -N₂ was solved in 1986 by Mills *et al.* (DOI: 10.1063/1.450310), and is known to be stable up to 60 GPa, pressure at which it transforms into ζ -N₂. While the lattice parameters of ϵ -N₂ are well established in the literature, its full structure refinement at pressures higher than about 20 GPa has not been done. As such, we have prepared an additional diamond anvil cell and obtained a full structure refinement of ϵ -N₂ at 54 GPa from single-crystal X-ray diffraction data. The refined structure parameters are shown in Table S1, and the pressure point was added to Figure 3. We see that the obtained unit cell volume is in very good agreement with the literature and, as seen in Table S1, the atomic positions are also very close to those found in the literature (DOI: 10.1063/1.450310). With this, the “structural details” of ϵ -N₂ are very well established before the phase transition to ζ -N₂.

Regarding ζ -N₂, we now present the evolution of its unit cell up to 110 GPa based on new single-crystal data as well as the re-analysis of data obtained from authors of Ref. 18 (Figure 3 and new Figure S7) Moreover, the DFT calculations provide in-depth information on the electronic density reorganization in ζ -N₂ as a function of pressure. These together give a thorough description of ζ -N₂'s evolution with pressure.

Without these experimental characterization, the implication of the ϵ - ζ N₂ phase transition as well as the formation of amorphous nitrogen is not convincing.

Please see our response above regarding the importance of the ϵ - to ζ -N₂ phase transition.

As for “[...] formation of amorphous nitrogen is not convincing.”, this behavior of nitrogen is very well established ever since the year 2000 (see DOI: 10.1103/PhysRevLett.85.1262 and DOI: 10.1103/PhysRevB.64.052103). Indeed, these studies demonstrated that nitrogen progressively becomes amorphous from about 150 GPa.

In this manuscript, we propose a possible transformation pathway from ζ -N₂ into the amorphous phase using DFT calculations. As specified in the manuscript's main document, these calculations are meant to provide a qualitative insight. However, we conclude that with the experimental determination of the structure of ζ -N₂, it is now possible to investigate the underlying mechanism computationally. Our assessment of the bond order and changes to charge density distribution between intra- and intermolecular space is thereby a first important step towards understanding the amorphization of nitrogen.

The DFT results also raise many questions. For instance, why are the DFT calculated molecular volumes much smaller than experimental values for 63 and 70 GPa?

The differences between the calculated and the single-crystal data are, opposite to the Reviewer's assessment, rather small, being of ~2.5%. Considering an estimated uncertainty of 3 GPa in the experimental pressure determination, the difference between the theoretical and experimental volumes is reduced to ~1%. The same order of magnitude of error is found for static calculations for ϵ -N₂, where we find differences of about 1-3%. This is an acceptable agreement, and in line with what is commonly found in the literature (*see DOI: 10.1088/0953-8984/13/4/302; DOI: 10.1039/d2dt03433f; DOI: 10.1021/acs.chemmater.2c00520; DOI: 10.1021/acs.chemmater.6b02593; DOI: 10.1103/PhysRevB.93.054101; DOI: 10.1021/acs.chemmater.6b04538; DOI: 10.1063/1.5091947*). Note also, the calculations in underestimate the experimental volumes, meaning that if thermal expansion were to be included, the volume gap would be reduced.

To avoid any potential confusion, it was added to the revised manuscript that, given the mild underestimation of the unit cell volume by calculations, the provided pressures for the electronic density overlap is likely lower than in reality.

Calculated static enthalpy difference between ζ - and ϵ -N₂ seems to suggest that ζ -N₂ is always more stable than ϵ -N₂ over a very wide pressure range, from 20 GPa to >110 GPa. This contradicts with experimental observations, and why?

The Reviewer is correct that ζ -N₂ shows a lower static enthalpy (*i.e.* at 0 K, no phonon contributions) over the full pressure range shown in Figure S11 (previously Figure S9). However, this does not contradict experimental observations. Indeed, Bini *et al.* (J. Chem. Phys. 112, 8522 (2000)), based on low temperature experiments (down to 25 K) on molecular nitrogen proposed the following formula for the temperature-dependent phase transition of ϵ -N₂ to ζ -N₂: $P(\text{GPa}) = 0.115 \cdot T(\text{K}) + 18.270$. This suggests a phase transition pressure of 18.27 GPa at 0 K. If we extrapolate our static enthalpy calculations of Figure S11 to lower pressures, we obtain a calculated transition pressure of ~10 GPa—in close agreement with the value reported by Bini *et al.*

This information is provided in our manuscript: “The static calculations are in qualitative agreement with experimental measurements which suggest the transition from ϵ -N₂ to ζ -N₂ to

occur 18.27 GPa at 0 K.²²” To ensure this statement is not overlooked, we have also added it to the caption of Figure S11.

A minor note: “The blue, pink, green and orange spheres correspond to the N1, N2, N3 and N4 atoms” in Fig. 2 and the text is not right. N3 atom should be orange based on authors’ description. We thank for Reviewer for pointing this out. This error was corrected.

In sum, the goal of the manuscript is scientifically important. But the study does not present sophisticated, convincing experimental or theoretical characterisation to support the conclusion. We agree with Reviewer #2 that our work is scientifically important and believe that our improved manuscript should address all of his/her concerns.

Report of Reviewer #3

Using laser-heated screw-type BX90 diamond anvil cells Laniel and coauthors synthesized and identified ζ -N₂ with XRD and Raman spectroscopy. They experimentally determined the crystal structure of ζ -N₂ at 63 and 70 GPa and theoretically extended the predicted structure to 200 GPa. The structure determination of ζ -N₂ and full explanation of the previously collected detailed Raman spectroscopic data with regards to the known number of lattice and vibron modes represent an important progress. The theoretical approach that treated ζ -N₂ and κ -N₂ as minor distortions of the ε -N₂ and focused on the general linear trend helped to get the big picture of progressive intra-to-intermolecular electronic delocalization and identification of ELF agglomeration occurring in molecular nitrogen. The paper is publishable in Nature Communications with the follow comment.

We thank the Reviewer for this very positive assessment of our work.

The central contribution and strength of the present work is the single-crystal XRD (SCXRD) of ζ -N₂ that distinguish itself from many previous works since 2006 that relied on powder XRD (pXRD) data. However, the present main text only shows powder data in Fig. 1, and leave SCXRD in the Supplemental Materials which only show the processed final answer in S1 and S2. Without any detailed data and discussion on the quality of the single crystals, it is difficult to assess the experiment.

In response to the Reviewer’s comment, we added further information on the single-crystal X-ray diffraction data and the analysis procedure. As described in answer to other Reviewers’ comments above, on top of the Crystal Information File (CIFs), the corresponding CheckCIF report and Tables S1 and S2, we are now providing slices of the reciprocal space of the ζ -N₂ crystals at 63, 70 and 86 GPa (Figure S3, Figure S4 and Figure S5) and that of ε -N₂ at 54 GPa (Figure S1). The

latter provide supporting evidence for the high-quality single crystals on which the analysis is based.

Moreover, a full description of the data collection and analysis method was added to the Methods section. It can also be found in response to the second Reviewer's first comment.

REVIEWERS' COMMENTS

Reviewer #1 (Remarks to the Author):

The authors have addressed all my concerns, and the manuscript is suitable for publication now.

Reviewer #2 (Remarks to the Author):

I appreciate the authors for performing an additional single crystal XRD experiment at 86 GPa and re-analysing more powder XRD data points from an earlier report. Now the revised Fig 3 is much more convincing in terms of volume evolution with pressure across the ϵ - ζ N2 phase transition. I also appreciate the authors' efforts in explaining the results between DFT and experiments. The scientific goal of this work is important, as I have already laid out in the last round of review. The additional experiments, analysis, and explanations resolved my previous concerns and I now would recommend its publication.

Reviewer #3 (Remarks to the Author):

The authors have answered all my questions. I support its publication in Nature Comm.